# Impact of Sarcopenia and Bone Mineral Density on Implant Failure after Dorsal Instrumentation in Patients with Osteoporotic Vertebral Fractures

**DOI:** 10.3390/medicina58060748

**Published:** 2022-05-31

**Authors:** Harald Krenzlin, Leon Schmidt, Dragan Jankovic, Carina Schulze, Marc A. Brockmann, Florian Ringel, Naureen Keric

**Affiliations:** 1Department of Neurosurgery, University Medical Center, 55131 Mainz, Germany; leon.schmidt@unimedizin-mainz.de (L.S.); dragan.jankovic@unimedizin-mainz.de (D.J.); schulzecarina88@aol.com (C.S.); florian.ringel@unimedizin-mainz.de (F.R.); naureen.keric@unimedizin-mainz.de (N.K.); 2Department of Neuroradiology, University Medical Center, 55131 Mainz, Germany; marc.brockmann@unimedizin-mainz.de

**Keywords:** sarcopenia, osteoporosis, spondylodesis, vertebral fracture, frailty

## Abstract

*Background and Objectives:* Age-related loss of bone and muscle mass are signs of frailty and are associated with an increased risk of falls and consecutive vertebral fractures. Management often necessitates fusion surgery. We determined the impacts of sarcopenia and bone density on implant failures (IFs) and complications in patients with spondylodesis due to osteoporotic vertebral fractures (OVFs). *Materials and Methods:* Patients diagnosed with an OVF according to the osteoporotic fracture classification (OF) undergoing spinal instrumentation surgery between 2011 and 2020 were included in our study. The skeletal muscle area (SMA) was measured at the third lumbar vertebra (L3) level using axial CT images. SMA z-scores were calculated for the optimal height and body mass index (BMI) adjustment (zSMA_HT_). The loss of muscle function was assessed via measurement of myosteatosis (skeletal muscle radiodensity, SMD) using axial CT scans. The bone mineral density (BMD) was determined at L3 in Hounsfield units (HU). *Results:* A total of 68 patients with OVFs underwent instrumentation in 244 segments (mean age 73.7 ± 7.9 years, 60.3% female). The median time of follow-up was 14.1 ± 15.5 months. Sarcopenia was detected in 28 patients (47.1%), myosteatosis in 45 patients (66.2%), and osteoporosis in 49 patients (72%). The presence of sarcopenia was independent of chronological age (*p* = 0.77) but correlated with BMI (*p* = 0.005). The zSMA_HT_ was significantly lower in patients suffering from an IF (*p* = 0.0092). Sarcopenia (OR 4.511, 95% CI 1.459–13.04, *p* = 0.0092) and osteoporosis (OR 9.50, 95% CI 1.497 to 104.7, *p* = 0.014) increased the likelihood of an IF. Using multivariate analysis revealed that the zSMA_HT_ (*p* = 0.0057) and BMD (*p* = 0.0041) were significantly related to IF occurrence. *Conclusion:* Herein, we established sarcopenic obesity as the main determinant for the occurrence of an IF after instrumentation for OVF. To a lesser degree, osteoporosis was associated with impaired implant longevity. Therefore, measuring the SMA and BMD using an axial CT of the lumbar spine might help to prevent an IF in spinal fusion surgery via early detection and treatment of sarcopenia and osteoporosis.

## 1. Introduction

Loss of bone and muscle with age are emblematic signs of frailty, undermining independence and quality of life [1,2]. As the number of elderly worldwide stands to double between 2017 and 2050, disorders contributing to higher morbidity and health costs create a substantial socioeconomic burden [3]. In a prospective cohort study, total annualized healthcare costs increase from 5707 USD among robust to 20,027 USD among frail patients in the USA [4]. The European Working Group on Sarcopenia in Older People (EWGSOP) defines sarcopenia as declines in muscle mass and function [5]. By incorporating low muscle strength into this equation, it is their intent that the clinical awareness of sarcopenia will improve. Low muscle bulk and poorer muscle quality have generally remained topics of research and overlooked in clinical practice [6,7]. Sarcopenia is now considered severe when combined with impaired physical performance, low muscle strength, and declines in overall mass/quality of muscle [5]. Sarcopenia is clinically detectable through the patient history, pertinent questionnaires (e.g., SARC-F), and standardized and targeted examination (e.g., grip strength, gait speed) [8,9,10]. Although not regularly available to primary caregivers, magnetic resonance imaging (MRI) and computed tomography (CT) are considered gold standard methods for the noninvasive quantification of muscle mass [11]. Unfortunately, a shared clinical and technical diagnostic dilemma is the lack of well-defined cut-off values for muscle-deficient states [5]. A CT measurement of skeletal muscle cross-sectional area (SMA) at the third lumbar (L3) vertebral level seems to correlate well with whole-body muscle determination [12,13]. Revised EWGSOP guidelines also specify muscle mass as a correlate of body size [5]. Reasonable means of adjustments therein may include division by height^2^, weight, or body mass index (BMI) [14,15]. In 2021, Derstine et al. analyzed various BMI-adjusted z-scores that were derived from height-adjusted skeletal muscle area measurements (SMA_HT2_ = SMA/height^2^ and SMA_HT_ = SMA/height) [15]. An unbiased assessment of SMA at L3 across a full range of body sizes was previously achieved using z(SMA_HT_) [15]. Changes in the micro- and macroarchitecture or composition of muscle are indicative of altered muscle quality [7]. Imaging tools, often CT or MRI, were used to assess such changes, particularly fatty infiltration of muscle [16]. Higher muscle fat content results in low attenuation (low muscle density) and is associated with a lower specific force, which is a muscle quality surrogate [17]. For differentiation between normal muscle and states of myosteatosis, the extent of intramuscular adipose tissue is determinable using Hounsfield units (HU) [18]. Osteoporosis is a systemic skeletal disease that is characterized by low bone mass, low bone mineral density (BMD), and microarchitectural bony deterioration [19,20,21]. The interdependence of bone and muscle has been increasingly acknowledged given their close anatomic and mechanical ties [22]. Both tissues engage in privileged exchanges via paracrine and endocrine signals [23]. Common pathways involving inflammatory cytokines and anabolic and catabolic metabolites (i.e., myokines, osteokines), and the mechanical interaction ongoing during physical activity may contribute to the loss of muscle and bone mass [24]. It has recently been shown that patients with decreased BMD and osteoporosis may be identified based on CT-derived bony HU determinations [25,26]. Osteopenia is defined as values of 100–160 HU, whereas values < 100 are considered osteoporotic [25,26,27,28]. The risk of osteoporotic vertebral fractures (OVFs) is increased in patients at HU < 160 [28,29]. Obesity and adipose tissue are linked to sarcopenic fatty muscle degeneration and bone health through various pathways [30]. In surgical patients, clinically observed sarcopenia and diminished bone mass are known predictors of postoperative complications, prolonged hospitalization, poor outcome, and mortality [31]. Some patients in whom obesity, sarcopenia, and osteoporosis coexist comprise a distinct subset that is classifiable as osteosarcopenic obesity [32]. Unlike one such affliction, it is suspected that this particular pathologic triad may actually worsen health outcomes [33]. In a cross-sectional study, the incidence of sarcopenia was higher in patients with an OVF, suggesting that sarcopenia might be a risk factor [34]. Further, sarcopenia is implicated in causing spinal sagittal imbalance due to insufficient compensation [35]. An OVF often necessitates fusion surgery [36]. Meta-analysis revealed that implant failure is considerably higher in osteoporotic spines than with normal bone mineral density [37]. Given the tight interrelation of sarcopenia and osteoporosis, the impact of deminished muscle mass on implant failure after spinal fusion surgery has so far been neglected.

It was the objective of the present study to determine the impact of the L3 z(SMA_HT_) as a surrogate parameter of muscle mass and BMDs (HU) on the occurrence of an IF after the instrumentation of an OVF.

## 2. Materials and Methods

### 2.1. Patients

This retrospective, single-center cohort study included all patients with an OVF who underwent surgical instrumentation at our hospital between 1 January 2011 and 31 December 2020. An OVF is defined as a low-energy fragility fracture and is described by the OF classification system. Exclusion criteria were treatment of an OVF with either vertebro- or kyphoplasty or insufficient imaging material. Baseline characteristics included age, sex, height (m), and weight (kg). According to World Health Organization International Classification standards, the BMI was calculated and categorized into groups. Comorbidities were assessed using the Charlson Comorbidity Index (CCI) [38,39]. The Groningen Frailty Index (GFI) and the G8 Questionnaire were used to evaluate patients according to their frailty and functional status [40]. All patients received dorsal instrumentation and, if necessary, cement augmentation. A routine follow-up CT was performed 24–72 h after surgery and after 3 to 12 months. Patients were encouraged to report immediately in the case of clinical deterioration. The study was split into different cohorts according to BMI (<25 and >25 kg/m^2^) and implant failure (IF; IF and no IF).

### 2.2. Surgical Complications and Implant Failure

Post-surgical complications and IFs were recorded during follow-up. Complications were defined as postoperative events with or without the need for radiographic, pharmacological, or surgical intervention. An IF was defined as no signal zone (halo) surrounding the whole screw in CT images and interpreted as screw loosening by radiologists [41,42].

### 2.3. Measurement of Sarcopenia and Osteoporosis

CT images were acquired using an ultra-high-resolution CT scanner (Aquilion 32 scanner; Toshiba-Canon Medical Systems, Neuss, Germany) with a matrix of 512 × 512 and a focal spot size of 1.6 × 1.4. Further applications included a detector element size of 0.5 mm and beam collimation of 0.5 × 32 mm rows prior to surgery. Plane imaging at the mid pedicle level of L3 was selected. The SMA determination included the psoas and the erector spinae muscle. The sum of the cross-sectional areas of all skeletal muscles was calculated, averaged, and divided by the height^2^ (SMA_HT_). The z(SMA_HT_) score was computed for each subject, with SMA_HT_ (cm^2^/m): *I* = *SMA_i_*/*height_i_*, i (sex = 1 if male, and 0 if female). The final z(SMA_HT_) equation was z = (I − I’)/SD(I), where I = SMA/height; I’ = 50 + BMI + 13 × sex + 0.6 × BMI × sex, SD (I) = 8.8 + 2.6 × sex [15]. The SMD (in HU) was automatically calculated using an unenhanced cross-sectional CT at the same level. According to EWGSOP standards, sarcopenia was defined as z(SMA_HT_) at L3 < −1 and a low SMD (men: <37.42 HU; women: <33.17 HU) [18,43] (Figure 1).

### 2.4. Statistics

Findings were reported as the mean or median ± SD. For the statistical analysis, we used Fisher’s exact test. Each group was tested for a Gaussian distribution if one-way ANOVA was passed, followed by Bonferroni’s test. If this failed, the Kruskal–Wallis test followed by Dunn’s correction was conducted to test for significance among multiple groups. Pearson’s correlation with nonlinear regression analysis was performed to compute Pearson’s r coefficient and *p*-values. Statistical analyses were performed using Microsoft Office Excel 2011 or Graph Pad Prism 8.4.2 for macOS, GraphPad Software, La Jolla, CA, USA. A value of *p* < 0.05 was accepted as statistically significant.

## 3. Results

### 3.1. Baseline Characteristics

Between 1 January 2011 and 31 December 2020, 68 patients with an OVF were treated using dorsal spinal instrumentation at our department. Cement augmentation was used to increase the stability of pedicle screws if the bone quality was poor. Patients treated with kypho- and vertebroplasty and those without sufficient imaging material for analysis were excluded from our analysis. Of all patients, 41 (60.3%) were female and 27 (39.7%) were male. The patients’ ages ranged from 56 to 86 years (73.7 ± 7.9 years). Sixty-eight patients with an OVF underwent instrumentation in 244 segments. The median time of follow-up was 14.14 ± 15.45 months. The comorbidity burden and estimated 10-year survival were measured using the CCI. The median CCI was 4 (range 2–6) and the GFI was 4 (range 2–6). Forty-four (64.7%) of all patients had a GFI score of 4 or greater, rendering them frail. Forty patients had a CCI of 4 or higher, limiting their 10-year survival probability to 53.0% and lower. Females were 1.61 ± 0.06 m tall with a BMI of 27 on average, while males were 1.74 m tall with a BMI of 27. Using axial CT, sarcopenia was detected in 28 patients (47.1%), myosteatosis in 45 patients (66.2%), and osteoporosis in 49 patients (72.0%) (Table 1). The majority of fractures were OF 4 with either a loss of integrity of the vertebral frame structure, vertebral body collapse, or pincer-type fracture (72.7%) (Table 2).

### 3.2. Postoperative Complications

Cement leakage occurred in 4 cases (out of a total of 28 cases with cement augmentation), hematoma in 1 case, and surgical site infection in 2 cases. In total, seven postoperative complications occurred in six patients (9.1%). Complications only occurred in patients with sarcopenia (*p* = 0.01).

### 3.3. Sarcopenia, BMI, and Osteoporosis

According to EWGSOP, 32 (47.1%) patients were classified as sarcopenic; 58 (85.3%) patients had an elevated BMI, while 14 (20.6%) were overweight and 44 (64.7%) were obese. Sarcopenia correlated significantly with elevated BMI (*p* = 0.005). The mean SMA was 6.3% lower in the “BMI < 25 kg/m^2^” compared with the “BMI > 25 kg/m^2^” cohort (*p* = 0.33). The mean zSMA_HT_ difference between both groups was −1.48 ± 0.72 (95% CI: −2.921 to −0.04736; *p* = 0.04). Further, lower zSMA_HT_ scores were associated with a lower BMD (*p* = 0.04). The degree of fatty muscle degeneration was significantly lower in the “BMI under 30” cohort (*p* = 0.03) (Figure 2).

### 3.4. Sarcopenia, BMD, and Implant Failure

The mean follow-up was 14 ± 15.5 months. Nineteen patients suffered from an IF during the follow-up time. An IF occurred in 14 (43.8%) patients with sarcopenia and five (14.7%) patients with normal muscle mass and function (*p* = 0.009). The mean SMA was 15.7% lower within the “IF group” compared with the “no IF group” (*p* = 0.005). The average zSMA_HT_ scores were significantly lower in those with an IF (−1.330) than in those without (−0.22). Both sarcopenia (OR 4.511, 95% CI 1.459–13.04, *p* = 0.0092) and osteoporosis (OR 9.50, 95% CI 1.497 to 104.7, *p* = 0.014) increased the likelihood of an IF. A multivariate logistic regression analysis was conducted to identify independent predictors of an IF in patients with an OVF. The zSMA_HT_ (*p* = 0.0057) and BMD (*p* = 0.0041) were independently related to an IF occurrence (Table 3). Cement augmentation of pedicle screws was associated with lower rates of IFs in patients with BMD < 80 HU (*p* = 0.003) but not in patients with sarcopenia (*p* = 0.1585) (Figure 2, Table 3).

### 3.5. Sarcopenia, Chronological Age, and Frailty

The presence of sarcopenia was independent from chronological age (*p* = 0.77), frailty (G8: *p* = 0.81; GFI: *p* = 0.056), and the presence of comorbidities (CCI: *p* = 0.77). Using multivariate logistic regression analysis, age and frailty did not correlate with the occurrence of IFs.

## 4. Discussion

Sarcopenia has long been associated with old age but is now believed to begin earlier in life, depending on the array of coexistent, contributory ailments other than aging [44]. In 2018, the EWGSOP published its revised definition of sarcopenia, and it is now formally accepted, bearing its own ICD-10 diagnosis code [24]. Their definition combines low muscle strength (being the most reliable measure of muscle function) with reductions in muscle mass and quality [24]. Impaired muscle quality was added to account for micro- and macroscopic changes in muscle architecture and composition. In a stepwise algorithm, sarcopenia is probable if low muscle strength is detected (e.g., using the grip strength test) [45], and it is confirmed by the added presence of a reduction in muscle mass or quality [5]. Physical performance (e.g., gait speed), formerly considered part of the core definition, is instead regarded as an outcome parameter to be used for grading disease severity [5]. For the first time, our data associated sarcopenic obesity and BMD with the occurrence of an IF after instrumentation. Here, the measurement of the SMA and BMD using axial CT of the lumbar spine was used to detect sarcopenia and osteoporosis.

Over the past decade, our understanding of the role that muscle plays in health and disease has advanced appreciably [46]. We now know that a higher body weight leads to fatty muscle infiltration (myosteatosis), exacerbating sarcopenia [47]. The combination of sarcopenia and obesity is currently seen as a distinct physical condition termed sarcopenic obesity [48]. Recently, the concept of a bone–muscle unit, based on the cross-talk between both tissues, was supported by numerous studies [49]. Secreted molecules released by the skeletal muscle affect the bony skeleton, just as osteokines act on adjacent muscle cells [23]. Despite the importance of these biochemical interactions, mechanical forces and gravity are still integral facets of the biomechanical model [50].

Sarcopenia and overweightness often coincide with osteoporosis in another novel entity: osteosarcopenic obesity [51]. The data we recorded were aligned with the abovementioned concepts, with more than half of all patients qualifying as overweight or obese, while half of them qualified as sarcopenic. Sarcopenia correlated significantly with an elevated BMI and lower BMD, further substantiating their interrelation. Lower SMA and sarcopenic states were not strictly tied to obesity but reflected overweightness and an increased BMI (>25 kg/m^2^). Sarcopenia thus occurred independently, unrelated to chronological age, furthering the notion that it is truly a disease and not merely a by-product of old age and frailty. The prevalence of sarcopenia in older-adult community dwellers seems to range from 0.8–26%, depending on the definition and diagnostic tools applied [52,53]. Previous studies also reported that 27.9% to 37.0% of patients with an OVF are sarcopenic [54]. A cohort study by Life Line Screening identified sarcopenia in 45.5% of overweight adults (BMI ≥ 25 kg/m^2^) and in 6.1% of obese adults (BMI ≥ 30 kg/m^2^) [55]. Our results are in agreement with these data, where the overall prevalence of sarcopenia was 46% and 85.3% of all overweight and obese patients, respectively. So far, sarcopenia is seen as a sign of old age reflecting frailty. Here, the prevalence of sarcopenia was independent of chronological age and frailty, as measured by frailty indices, and only depending on BMI. One possible explanation might be the inherent selection bias that occurred by selecting patients with an OVF treated using dorsal instrumentation. This constituted a highly selected subgroup of patients. However, our findings might well be extrapolated within the field of spinal surgery.

Owing to technological limitations, muscle mass and muscle quality may be difficult to assess as primary parameters in some clinical settings [7]. L3 SMA readings using CT or MRI are the established gold standard for measuring muscle mass and quality [15], correlating significantly with whole-body muscle content [13]. The z(SMA_HT_) score used herein afforded optimal height and BMI adjustments of L3 SMA values [13].

Clinically, sarcopenia increases the risk of falls and fractures and impairs activities of daily life [56]. The association of sarcopenia with cognitive impairment and cardiac or respiratory disease has been documented for some time [57,58,59]. However, its importance as an independent prognostic factor for gauging therapeutic responses and overall mortality has fueled research interests more recently [60]. As many aspects of epidemiology and pathophysiology are better understood, links between muscle pathology and adverse health outcomes have surfaced [24]. Sarcopenia was studied in conjunction with various malignancies, such as advanced oesophageal cancer, gastrointestinal malignancies, melanoma, and lung cancer [60,61,62,63], and in relation to various therapies, including radiation, immune-checkpoint inhibition, and chemotherapy [61,64]. The prognostic implications of sarcopenia for malignant disease or spinal disorders are indisputable. In older patients, chronic lower back pain is strongly associated with coexistent sarcopenia [65]. In cases of lumbar spinal stenosis, sarcopenia leads to more severe low back pain, a greater degree of slippage, and impaired physical function [66]. Sarcopenia-induced changes in sagittal balance, as measured by pelvic tilt, may explain how spinal disorders and loss of muscle mass are interrelated [67]. In the presence of sarcopenia, lower Japanese Orthopedic Association scores and recovery rates were observed following spinal surgery [68].

Thus far, little is known of sarcopenia as a prognostic factor of an IF after spinal instrumentation and fusion surgery in patients with osteoporosis. In the present study, postoperative complications (9.3%) were confined to those with sarcopenia. Similar outcomes have resulted after oncologic gastrointestinal surgery, the risk of major postoperative complications and events in total were worsened by sarcopenia [69]. Likewise, sarcopenia was shown to be an independent risk factor for postoperative infectious complications in patients with gastric cancer [70]. In our study, IFs occurred significantly more often in patients with sarcopenia, and both sarcopenia and lower BMD emerged as independent risk factors for an IF. For the first time, we established a link between diminished muscle mass and the occurrence of IFs after spinal instrumentation surgery. IFs occurred in 43.8% of patients with sarcopenia and in 14.7% of those with normal muscle mass. These results are consistent with reported rates of screw loosing after spinal fusion surgery of up to 15% in non-osteoporotic patients and even higher in those with a lower BMD [71]. It is particularly interesting that the number of implant failures in patients with sarcopenia equals the rates reported in patients with osteoporosis. This might further strengthen the tight interrelation and shared pathways leading to reduced muscle mass and bone mineral density. This is of particular interest because sarcopenia and osteoporosis are potentially treatable conditions [72,73]. Although supplemental measures, such as cement augmentation, helped with reducing IFs in patients with low BMDs, they did not impact those with sarcopenia.

On the other hand, implant failures after instrumentation were entirely unaffected by BMI. Earlier efforts achieved comparable results, citing no increases in minor complications after spinal deformity surgery in obese patients [74]. Obesity itself is therefore no impediment to spinal fusion surgery, as the long-term outcomes appear similar regardless of body weight [75].

Limitations of the study included its retrospective design with its inherent drawbacks. It is possible that determining osteoporosis based on axial CT by measuring BMD rather than standardized DXA measurement impaired the assessment of osteoporosis, and thus, the impact of osteoporosis on implant failure in our cohort. More definitive answers can only be obtained from systematic perspective registries. It was shown that sagittal balance, pedicle screw size and positioning, and biomechanics affect the rate of pedicle screw loosening. These parameters were not addressed in this study. While not being the only contributing factors, sarcopenia and BMD were found to be associated with implant failure. Further studies are warranted to determine the importance of those in the delicate interplay that is spinal fusion surgery.

## 5. Conclusions

Our results showed for the first time that sarcopenia and obesity were mainly, and to a lesser extent osteoporosis itself, responsible for the occurrence of an IF after instrumentation of an OVF. Population aging has been recognized as one of the four global demographic “megatrends” affecting healthcare services worldwide. As the number of elderly increases, disorders contributing to higher morbidity and health costs create a substantial socioeconomic burden. Therefore, the measurement of SMA and BMD using axial CT of the lumbar spine may help reduce the incidence of IFs during spinal fusion surgery via early detection and treatment of sarcopenia and osteoporosis, which will ultimately lessen the detrimental impact of frailty on our society.

## Figures and Tables

**Figure 1 medicina-58-00748-f001:**
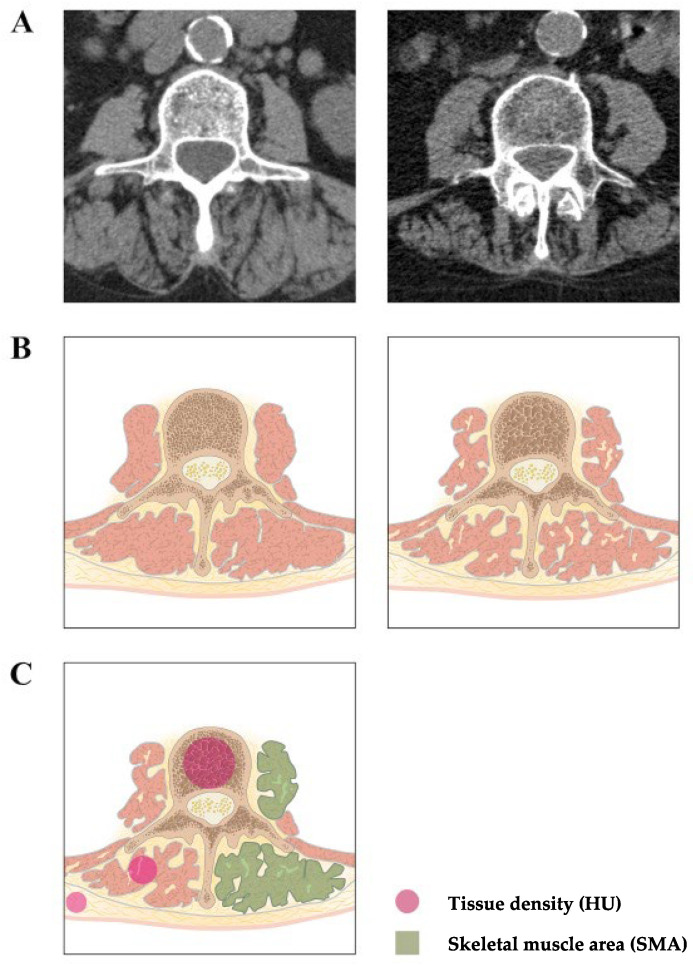
Plane imaging at the mid-pedicle level of L3 of patients with normal muscle mass (left) and sarcopenia (right). Computed tomography images(**A**), outlines (**B**), and skeleton muscle area (SMA) and skeleton muscle density (SMD) (in Hounsfield unit (HU)) measurements (**C**).

**Figure 2 medicina-58-00748-f002:**
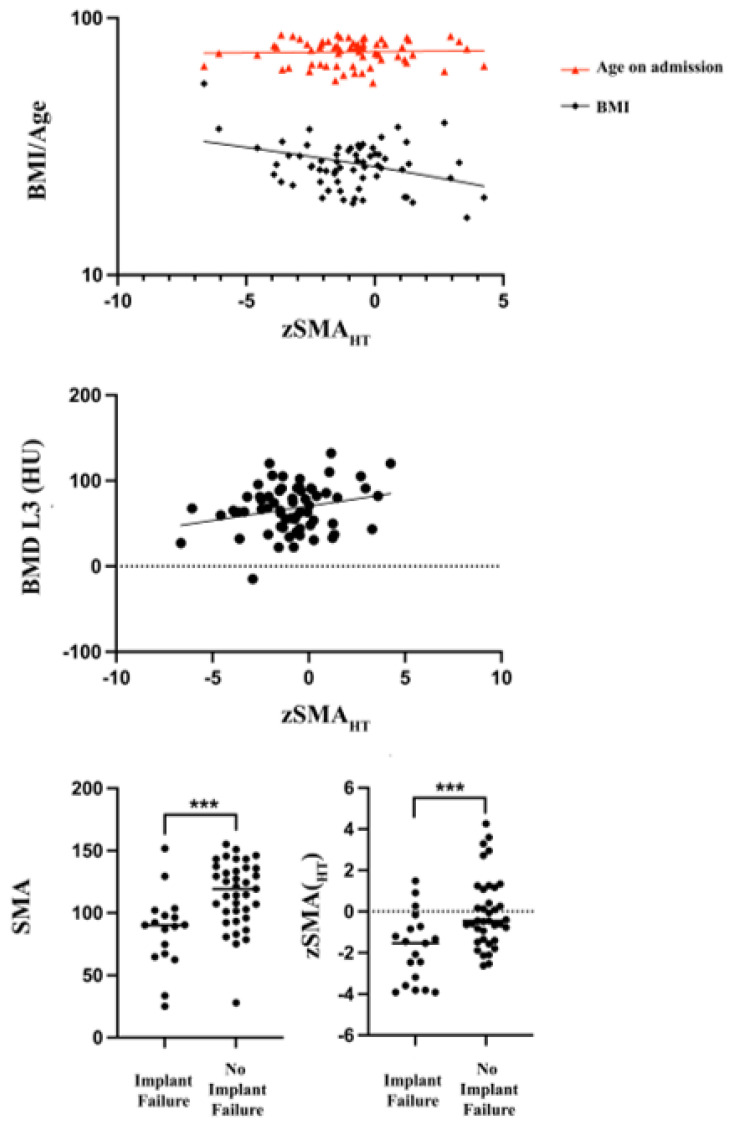
Correlation of zSMA_HT_ with age/BMI (top) and BMD L3 (middle). Association of SMA (lower left) and zSMA_HT_ (lower right) with implant failure. *** *p* ≤ 0.001.

**Table 1 medicina-58-00748-t001:** Baseline demographics. Charlson Comorbidity Index (CCI), Groningen Frailty Index (GFI), and body mass index (BMI).

	Total	No Implant Failure	Implant Failure
Age, mean (range)	73.7 (59–89)	73.4 (59–88)	72.5 (59–87)
Sex (%)			
Female	41 (60.3)	29 (59.2)	12 (63.2)
Male	27 (39.7)	20 (40.8)	7 (36.8)
Comorbidity and Frailty, median (range)			
CCI	4 (2–6)	4 (2–6)	4 (2–6)
GFI	4 (2–6)	4 (2–6)	4 (2–6)
Weight (kg), mean (SD)	76.1 (±20.9)	77.3 (±22.6)	73.1 (±15.9)
BMI (kg/m^2^), mean (SD)	25.1 (±6.9)	27.8 (±6.6)	27.2 (±5.8)
Localization instrumentation			
Thoracic	11 (16.7)	8 (16)	3 (18.8)
Thoracolumbar	39 (59.1)	31 (62)	8 (50)
Lumbar/lumbosacral	11 (16.7)	8 (16)	3 (18.8)
Lumbosacral	5 (7.6)	3 (6)	2 (12.5)
Instrumented levels			
1 level	0	0	0
2 levels	13 (19)	8 (16)	5 (31.3)
3 levels	3 (4.5)	2 (4)	1 (6.3)
4 levels	41 (62.1)	33 (66)	8 (50)
5 or more levels	9 (13.6)	7 (14)	2 (12.5)

**Table 2 medicina-58-00748-t002:** Osteoporotic fracture classification (OF).

	Total	No Implant Failure	Implant Failure
1	0	0	0
2	5 (7.6)	2 (4)	3 (18.8)
3	7 (10.6)	6 (12)	1 (6.3)
4	48 (72.7)	37 (74)	11 (68.8)
5	6 (9.1)	5 (10)	1 (6.3)

**Table 3 medicina-58-00748-t003:** Results of skeletal muscle area (SMA), bone mineral density (BMD) and implant failure. * *p* ≤ 0.05, *** *p* ≤ 0.001.

	Total	No Implant Failure	Implant Failure
SMA, mean (SD	105.9 (±31)	115.2 (±26.7) ***	85.8 (±30.7) ***
zSMA(_HT_), mean (SD)	−0.6 (±0.3)	0.1 (±0.3) ***	−1.8 (±0.4) ***
BMD	77.5 (±4.9)	81 (±6.5) *	65 (±4.3) *

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
