# Peer review of "Impact of Sarcopenia and Bone Mineral Density on Implant Failure after Dorsal Instrumentation in Patients with Osteoporotic Vertebral Fractures"

_medicina, 2022, doi:10.3390/medicina58060748_

Round 1

Reviewer 1 Report

1) Abstract. Age-related loss of bone and muscle mass are signs of frailty and associated with increased risk of falls and consecutive vertebral fractures. Management often necessitates fusion surgery. We determine the impact of sarcopenia and bone density on implant failure (IF) and complications in patients with spondylodesis due to osteoporotic vertebral fractures (OVF). Please add the name of section: “Background”.

2) Abstract. Conclusion: To a lesser degree, osteoporosis was associated with impaired implant longevity. Please underline the results that support this seclusion.

3) 1. Introduction L34-35. Age-related losses of bone and muscle are emblamatic of frailty, serving to under- mine independence and quality of life.[1]. Please improve this paragraph and add this reference:

  1. a) Prevalence of Sarcopenia Among the Elderly in Korea: A Meta-Analysis. J Prev Med Public Health. 2021;54(2):96-102. doi:10.3961/jpmph.21.046

4)1. Introduction. L65-66. Osteoporosis is a systemic skeletal disease, characterized by low bone mass, low bone marrow density (BMD), and microarchitectural bony deterioration.[17]. Please improve this references and add these references:

  1. a) Use of Trabecular Bone Score (TBS) as a Complementary Approach to Dual-energy X-ray Absorptiometry (DXA) for Fracture Risk Assessment in Clinical Practice. J Clin Densitom. 2017;20(3):334-345. doi:10.1016/j.jocd.2017.06.019
  2. b) Correlation between bone quality and microvascular damage in systemic sclerosis patients. Rheumatology (Oxford). 2018;57(9):1548-1554. doi:10.1093/rheumatology/key130

5) 1.Introduction. L90-92. In the present study, L3 SMA served as a surrogate parameter of muscle mass, using the ratio of subcutaneous fat and muscle tissue density to indicate muscle quality. To  measure BMDs, we obtained HU readings of L3 vertebrae. Please ameliorate the description of study aim.

6) 3. Results L139-145. 3.1. Baseline characteristics. Please underline the treatment regime of patients.

7) 4. Discussion L187-190. Sarcopenia has long been associated with old age but is now believed to begin earlier  in life, depending on the array of coexistent, contributory ailments other than aging.[37] In 2018, the EWGSOP’s published its revised definition of sarcopenia, and it is now formally accepted, bearing its own ICD-10 diagnosis code.[20] Please summarise here the most important results of the study.

8) 5. Conclusions L286-290. Our results show for the first time that sarcopenia and obesity are mainly, and to a  lesser extent osteoporosis itself, responsible for the occurrence of IF after instrumentation  of OVF. Measurement of SMA and BMD by axial CT of the lumbar spine may help reduce  the incidence of IF during spinal fusion surgery by early detection and treatment of sarcopenia and osteoporosis. Please underline the clinical implication of your article.

Author Response

We would like to express our gratitude for the appreciation of our manuscript. In order to comply with the conveyed suggestions for improvement, we have extensively revised our manuscript. For detailed answers please see the attachment.

Reviewer 2 Report

This study determined the impact of sarcopenia and osteoporosis on implant failure in patients with vertebral fracture

Abstract:

explain the abbreviation OF

In the abstract you state that osteoporosis predicts implants failure to a lesser degree than sarcopenia; however, the odds ratio for osteoporosis is higher than that of sarcopenia

Lines 38 and 39: it is stated that the healthcare costs associated with sarcopenia in the US is estimated to be $18.5 billion. the reference for this statement is not the original study. Also, this cost number is quite old; it is about 20 years old. Please update the cost or state that this is the cost from 20 years ago.

Line 46: I am not sure if that “converge” is the correct word to use here; please consider revising.

Line 65 replace the word allowing with allow

Line 67 I think “bone marrow density”should be “bone mineral density”

Line 69 I am not sure that “privileged exchanges” is the correct wording to use here; please consider re-wording

Line 91 it is indicated that the ratio of subcutaneous fat to muscle density was used to determine muscle quality. please check that this is correct

In the introduction you need to explain the abbreviation OVF

Please include a hypothesis statement at the end of the introduction

Line 98: Please define the abbreviation “OF”

Line 101-104: please include references for these questionnaires and classification systems

Line 107 please justify the cut off value of 20 that you have used for BMI

Line 124 please explain the abbreviation SMD

Please explain all the abbreviations for figure 1 legend

In figure 1: should smooth muscle area be skeletal muscle area?

Line 147: change to “The median time of follow-up was…”

Line 148: “Median CCI was 4 (range 2 - 6), and the GFI was 4 (range 2 - 6).” Please provide a brief interpretation of these results here for readers who are not familiar with these scales (at least for CCI; you’ve interpreted GFI on the next line).

Line 153: Please indicate or interpret what “OF 4” means.

Table 1: please include units

Table 1 and 2: Define abbreviations used in the tables in a footnote to the table

Lines 159-160: “Complications only occurred in patients with sarcopenia (p=0.01)”. Please indicate how this was statistically analyzed.

Line 163: Change “overweighted” to “overweight”

Lines 164: “Sarcopenia correlated significantly with elevated BMI (p=0.005)” – should this be “Sarcopenia correlated significantly with lower BMI (p=0.005)”?

Line 169: The reference to Figure 1 should be Figure 2 here.

Figure 2 legend: Please define all abbreviations in the legend. Please indicate in the legend what the “***” means in the bottom figure panel. Please also indicate the R (correlation) and whether these were significant.

Line 173: Again, please indicate how this statistical comparison between percentages for the groups was made.

Line 179: Here you refer to Table 2 for the multiple regression results, but this is not presented in Table 2.

Line 181: Reference is made to figure 2 here, but this figure does not correspond to the results that are stated here.

Table 3: I don’t think you have referred to this table in any part of the text.

Table 3: Please provide units in this table. Also, define the abbreviations in a footnote. Please also explain what the “*”, “**”, “***” mean in a footnote.

Line 225: Change “overweighted” to “overweight”

Line 269: Change “particular” to “particularly”

Line 270: Change “equels” to “equals”

Line 279: Change “similarly” to “similar”

Author Response

(The authors gave the same response as above.)
